# Vertical transmission of maternal DNA through extracellular vesicles associates with altered embryo bioenergetics during the periconception period

David Bolumar[1†], Javier Moncayo-Arlandi[2†], Javier Gonzalez-Fernandez[2], Ana Ochando[2], Inmaculada Moreno[2], Ana Monteagudo-Sanchez[2], Carlos Marin[1], Antonio Diez[1], Paula Fabra[3], Miguel Angel Checa[3,4], Juan Jose Espinos[3,5], David K Gardner[6,7], Carlos Simon[2,8,9]*, Felipe Vilella[2]*

[1]Igenomix Foundation, INCLIVA Health Research Institute, Valencia, Spain; [2]Carlos Simon Foundation, INCLIVA Health Research Institute, Valencia, Spain; [3]Clinica Fertty, Barcelona, Spain; [4]Department of Medicine and Life Sciences, University Pompeu Fabra, Barcelona, Spain; [5]Department of Pediatrics, Obstetrics and Gynecology, School of Medicine, UAB, Bellaterra, Spain; [6]School of Biosciences, University of Melbourne, Parkville, Australia; [7]Melbourne IVF, East Melbourne, Australia; [8]Department of Pediatrics, Obstetrics and Gynecology, School of Medicine, University of Valencia, Valencia, Spain; [9]Department of Obstetrics and Gynecology, BIDMC, Harvard University, Boston, United States

*For correspondence:
csimon@fundacioncarlossimon.
com (CS);
fvilella@fundacioncarlossimon.
com (FV)

†These authors contributed equally to this work

Competing interest: The authors declare that no competing interests exist.

**Abstract** The transmission of DNA through extracellular vesicles (EVs) represents a novel genetic material transfer mechanism that may impact genome evolution and tumorigenesis. We aimed to investigate the potential for vertical DNA transmission within maternal endometrial EVs to the pre-implantation embryo and describe any effect on embryo bioenergetics. We discovered that the human endometrium secretes all three general subtypes of EV - apoptotic bodies (ABs), microvesicles (MVs), and exosomes (EXOs) - into the human endometrial fluid (EF) within the uterine cavity. EVs become uniformly secreted into the EF during the menstrual cycle, with the proportion of different EV populations remaining constant; however, MVs contain significantly higher levels of mitochondrial (mt)DNA than ABs or EXOs. During the window of implantation, MVs contain an eleven-fold higher level of mtDNA when compared to cells-of-origin within the receptive endometrium, which possesses a lower mtDNA content and displays the upregulated expression of mitophagy-related genes. Furthermore, we demonstrate the internalization of EV-derived nuclear-encoded (n)DNA/mtDNA by trophoblast cells of murine embryos, which associates with a reduction in mitochondrial respiration and ATP production. These findings suggest that the maternal endometrium suffers a reduction in mtDNA content during the preconceptional period, that nDNA/mtDNA become packaged into secreted EVs that the embryo uptakes, and that the transfer of DNA to the embryo within EVs occurs alongside the modulation of bioenergetics during implantation.

## eLife assessment

This manuscript reports **important** results on the potential influence of maternally derived extracellular vesicles on embryo metabolism. The study combines **convincing** techniques for isolating different subtypes of EV, DNA sequencing, embryo culture, and respiration assays performed on human endometrial samples and mouse embryos. These findings set the stage for in-depth studies

to elucidate the role of EV contents in embryo energetics and further enhance our understanding on maternal-fetal communication during peri-implantation development.

## Introduction

The release and uptake of membrane-enclosed compartments with specific cargos, commonly known as extracellular vesicles (EVs), represents a critical cell-to-cell communication mechanism (*van Niel et al., 2018*). EVs support the transport of molecules and their protection from the extracellular environment (*van Niel et al., 2018*) under both physiological and pathological conditions (*Simon et al., 2018*). EVs are generally classified into three populations based on their biogenic pathways, composition, and physical characteristics: apoptotic bodies (ABs), microvesicles (MVs), and exosomes (EXOs) (*EL Andaloussi et al., 2013*). All EV subtypes protect and transport nucleic acids, including all known types of RNA (*van Balkom et al., 2015*; *Huang et al., 2013*; *Valadi et al., 2007*; *Vojtech et al., 2014*) and DNA (*Lázaro-Ibáñez et al., 2014*; *Thakur et al., 2014*; *Waldenström et al., 2012*).

Endometrial EVs secreted into the endometrial fluid (EF) participate in embryo development (*Burns et al., 2014*; *Burns et al., 2016*; *Ruiz-González et al., 2015*), embryonic implantation (*Greening et al., 2016*; *Ng et al., 2013*; *Vilella et al., 2015*), trophoblast migration (*Desrochers et al., 2016*), and endometrial angiogenesis (*Bidarimath et al., 2017*; *Salomon et al., 2014*) during the periconceptional period. Micro(mi)RNAs represent the human endometrium's most widely studied EV cargo. The Salomonsen group first identified miRNAs as a cargo of endometrial epithelium-secreted EXOs, with the contained miRNAs found to target genes involved in embryonic implantation (*Ng et al., 2013*). Our studies revealed that the human endometrium secretes EXOs containing miRNAs with distinct menstrual cycle phase-related profiles into the EF (*Vilella et al., 2015*). We discovered that miR-30d uptake by the embryo promotes the expression of genes encoding factors involved in embryo adhesion. Additional studies suggest that both the embryo and the maternal endometrium release specific sets of miRNAs with transcriptional/epigenetic-modifying potential that participate in embryo viability, implantation, and uterus preparation during the preconception period (*Ashary et al., 2018*; *Gross et al., 2017*; *Liang et al., 2017*).

Evidence regarding DNA transmission through EVs also suggests that specific sorting of DNA molecules may occur depending on the cell type, EV subpopulation (*Lázaro-Ibáñez et al., 2014*), and ability of EVs to carry DNA to target cells (*Waldenström et al., 2012*). EVs contain single- and double-stranded DNA in varying relative abundances depending on the cell and vesicle type (*Thakur et al., 2014*). The vertical transmission of EV-derived DNA has been proposed as a novel genetic material transfer mechanism that impacts genome evolution (*Kawamura et al., 2017*). In support of this proposal, studies have indicated the involvement of the horizontal transfer of EV-derived DNA (especially involving retrotransposons) in tumorigenesis (*Kawamura et al., 2017*). Interestingly, EVs also transport mitochondrial (mt)DNA, constituting a mechanism to transmit normal or mutant mtDNA associated with specific pathologies from cell to cell (*Guescini et al., 2010*). For example, T-cell-derived EXOs secreted upon antigen-dependent contact with dendritic cells can transfer nuclear-encoded (n)DNA and mtDNA unidirectionally to dendritic cells, triggering resistance to subsequent viral infections (*Torralba et al., 2018*).

Here, we investigated the vertical transmission of DNA encapsulated within EVs secreted from cells of the endometrium to the embryo and explored the possible consequences of this process. Our data demonstrate that all EV subtypes encapsulate nDNA and mtDNA and that the endometrium releases MVs that specifically encapsulate an elevated amount of mtDNA during the receptive phase of the menstrual cycle. Embryos display evidence of EV-derived DNA uptake during the periconceptional period, which associates with an increased metabolic rate of the embryos and suggests a role in bioenergetic modulation.

## Results

### Morphological, molecular, and nanoparticle tracking-mediated identification of endometrial EV populations secreted into the endometrial fluid

We first isolated EV subpopulations from EF samples obtained at distinct time points during the female menstrual cycle to investigate those EVs secreted by the endometrium. We analyzed EVs using transmission electron microscopy (TEM), protein marker expression, dynamic light scattering (DLS), and nanoparticle tracking analysis (NTA).

During the receptive phase (phase IV of the natural menstrual cycle), we identified the existence of ABs from EF samples in sizes ranging from 1.5 μm to 8 μm (*Figure 1A* and *Figure 1—figure supplement 1A and B*), which appeared as a multimodal population of two main sizes in DLS analysis (Note: NTA could not be used to measure AB concentration due to their large size). A main AB population displayed a mean size of 2029 nm and accounted for 61.6% of total measured particles, while a second population displaying a mean size of 274.7 nm accounted for 38.4% of total particle content (*Figure 1B*). ABs expressed all AB/MB molecular marker proteins evaluated (i.e. ARF6, VDAC1, Calreticulin, Calnexin, TSG101, CD63, and CD9) due to their heterogeneous origin (*Figure 1C*), while the detection of membranous-like structures within ABs (*Figure 1—figure supplement 1B*, Image 3) suggested that this EV population also encapsulated different organelle structures. While these structures may represent mitochondria, we could not study their presence with organelle-specific markers.

MVs displayed sizes ranging from 200 to 700 nm and possessed a heterogeneous electron-dense content, as noted by TEM (*Figure 1D* and *Figure 1—figure supplement 1C*, Image 3). MVs existed as a single population (98.7%) with a mean size of 290.8 nm by DLS (*Figure 1E*) and expressed specific molecular marker proteins (i.e. Calnexin, Calreticulin, ARF6, CD9, CD63, TSG101, and VDAC1) (*Figure 1F*). We observed an average MV concentration of $3.27·10^9±6.22·10^8$ part/mL, as analyzed by NTA (*Figure 1—figure supplement 2A*).

TEM demonstrated the existence of homogeneously-structured EXOs in sizes ranging from 40 to 160 nm (*Figure 1G* and *Figure 1—figure supplement 1D*). EXOs existed as a single-sized population (95.8%) (partially overlapping with MVs) with a mean size of 143.2 nm by DLS (*Figure 1H*) and expressed the EV-specific markers CD9, CD63, and TSG101 (*Figure 1I*). Finally, EXOs displayed a similar abundance as MVs ($3.18·10^9±2.31·10^8$ part/mL), as analyzed by NTA (*Figure 1—figure supplement 2B*).

We also measured the relative abundances of secreted MVs and EXOs across the menstrual cycle (*Figure 1J and K*) (as mentioned before, ABs cannot be measured using NTA). MV concentrations remained similar throughout the menstrual cycle (*Figure 1J*), ranging from minimal levels in the post-receptive phase (phase V) ($5.26·10^7±1.65·10^7$ part/mL) to maximal in the proliferative phase (phase I) ($3.48·10^8±2.03·10^8$ part/mL) (*Figure 1—figure supplement 3A*). EXO concentrations displayed greater variability throughout the menstrual cycle, although these changes remained non-significant (*Figure 1K*); overall, the number of particles secreted per ml did not significantly change (*Figure 1—figure supplement 3B*).

In summary, we discovered that the human endometrium secretes the three major EV subtypes and that the concentration of MVs and EXOs secreted into the EF does not significantly change during the different phases of the menstrual cycle.

### Sequencing of endometrial EV-derived DNA isolated during the periconception period

Next, we sequenced the DNA content of EF-derived EV populations (n=10) isolated during the periconceptional period (receptive endometrium, phase IV). We first used DNase pre-treatment of EVs to ensure bona fide DNA content for sequencing (*Figure 2—figure supplement 1A and B* and Materials and Methods). We compared the coding sequences obtained from isolated EVs with the ENSEMBL database, revealing the homogenous and more differentiated nature of ABs compared to MVs and EXOs, which possessed a more dispersed overlapping distribution. The grouping together of AB samples indicates a similar content, which differs from other EV subtypes. ABs, mainly generated by apoptotic cells, typically display greater homogeneity than MVs or EXOs, given their large size and

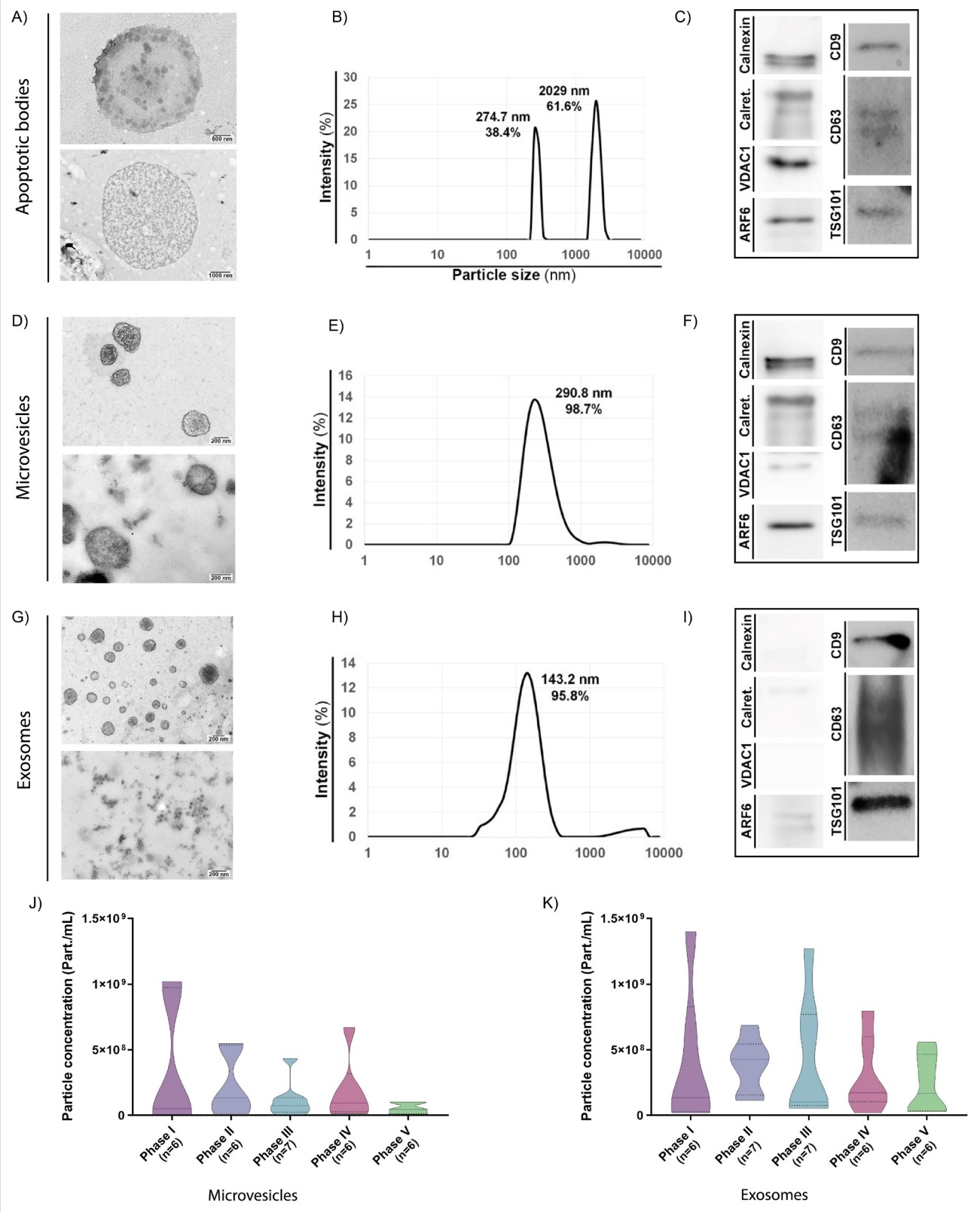

**Figure 1.** Characterization of endometrial fluid-derived extracellular vesicles. (**A–I**) Analysis of ABs, MVs, and EXOs isolated from human EF samples: morphology by TEM (**A**, **D**, and **G**), size distribution by DLS (**B**, **E**, and **H**), and protein marker expression by Western blotting (**C**, **F**, and **I**). TEM images obtained using two different protocols for an external (deposition processing, upper images) or internal (ultrathin slide processing, lower images) view of EVs. Size distribution analyzed in a single EF sample by DLS during the receptive phase for (**B**) ABs, (**E**) MVs, and (**H**) EXOs. Graphs show the

*Figure 1 continued on next page*

*Figure 1 continued*

average size distribution and percentage of total particles contained within the populations. Specific protein markers analyzed by Western blotting for (**C**) ABs, (**F**) MVs, and (**I**) EXOs. Analyzed markers (and associated molecular mass) were calnexin (90–100 kDa), calreticulin (60 kDa), VDAC1 (31 kDa), ARF6 (18 kDa), CD9 (24 kDa), CD63 (30–60 kDa), and TSG101 (45–50 kDa). (**J and K**) Particle concentration and size distribution measured by NTA for (**J**) MVs and (**K**) EXOs secreted throughout the menstrual cycle. One-way ANOVA and Kruskal-Wallis rank sum tests performed to compare the distinct menstrual cycle phases - no significant differences were observed.

The online version of this article includes the following source data and figure supplement(s) for figure 1:

**Source data 1.** Unedited blots for *Figure 1C,F, and I*.

**Figure supplement 1.** Transmission electron micrograph analysis of human endometrial fluid-derived extracellular vesicle morphology.

**Figure supplement 2.** Size distribution of human endometrial fluid-derived microvesicles and exosomes measured by nanoparticle tracking analysis.

**Figure supplement 3.** Microvesicle and exosome dynamics in endometrial fluid samples isolated during the menstrual cycle.

the encapsulation of a greater amount of intracellular material, making their contents less specific (*Figure 2—figure supplement 1C*).

All EV subtypes can encapsulate DNA, and we analyzed if any differences existed regarding DNA content using paired comparisons, with results displayed as volcano plots. MVs displayed significant differences regarding DNA content compared to ABs (*Figure 2A*) and EXOs (*Figure 2B*), while ABs and EXOs displayed no significant differences (data not shown). This suggests that MVs contain more specific DNA sequences than other EVs, as shown in volcano plots regarding FWER ($\alpha < 0.05$) and >twofold-change.

While elucidating the specific sequences of DNA encapsulated within EVs, we observed an average $11.12 \pm 0.53$ fold change enrichment in thirteen specific known mitochondrial genes in MVs compared with EXOs (*Figure 2C*). MVs also displayed enrichment for two mitochondrial pseudogenes and a long intergenic non-coding RNA from the Chr1:536816–659930 genomic region compared to EXOs, and a mitochondrial pseudogene and three protein-coding genes from different genomic loci compared to ABs (*Figure 2C*).

These results demonstrate that all EVs generally encapsulate DNA (nDNA and mtDNA), although MVs contain higher levels of mtDNA than ABs and EXOs.

## Endometrial EVs encapsulate mtDNA while mtDNA levels become reduced in the receptive endometrium

To understand the encapsulation of mtDNA in EVs and their secretion by endometrial cells, we analyzed the relative mtDNA content compared to the nDNA content of endometrial tissue biopsies from women undergoing hormone replacement therapy (HRT) during the receptive or periconceptional period (*P*+5). We analyzed receptivity status using the endometrial receptivity analysis (ERA) test (n=70), which demonstrated the number of women in pre-receptive, receptive or periconceptional, and post-receptive periods. We quantified the ratio between mtDNA/nDNA content using qRT-PCR, as previously described (*DiezJuan et al., 2015*). We observed a significant reduction (both p<0.001) of mtDNA in receptive (n=25) and post-receptive endometria (n=23) compared to the pre-receptive (n=22) phase (*Figure 3A*); furthermore, the post-receptive endometrium displays a significantly lower content of mtDNA compared to the receptive endometrium (p<0.001).

We then sought to understand the mechanisms underlying the reduction of mtDNA levels in the periconceptional/receptive endometrium using the same HRT patient samples. Previous research has linked the activation of the cellular mitochondrial degradation and recycling system (autophagy) to the encapsulation of mtDNA into EVs and their subsequent release into the extracellular space (*Kumar et al., 2015*). Therefore, we investigated the expression of genes that regulate specific pathways involved in mitochondrial autophagy (mitophagy) and engulfment into autophagosome for degradation in the pre-receptive, receptive, and post-receptive endometrium (*Youle and Narendra, 2011*). We found that vesicular targeting receptors Sequestosome-1 (*SQSTM1*) and Microtubule-associated proteins 1 A/1B light chain 3 A (*MAP1LC3A*) and the mitophagy inductor PTEN-induced kinase 1 (*PINK1*) became significantly upregulated in the receptive and post-receptive stages when compared to the pre-receptive stage (p<0.001; *Figure 3B*). We also observed the significantly downregulated expression of constitutive components of the mitochondria, such as the ribosomal protein S16 (*MRPS16*), mitochondrial ribosomal protein S9 (*MRPS9*), and mitochondrial import receptor subunit

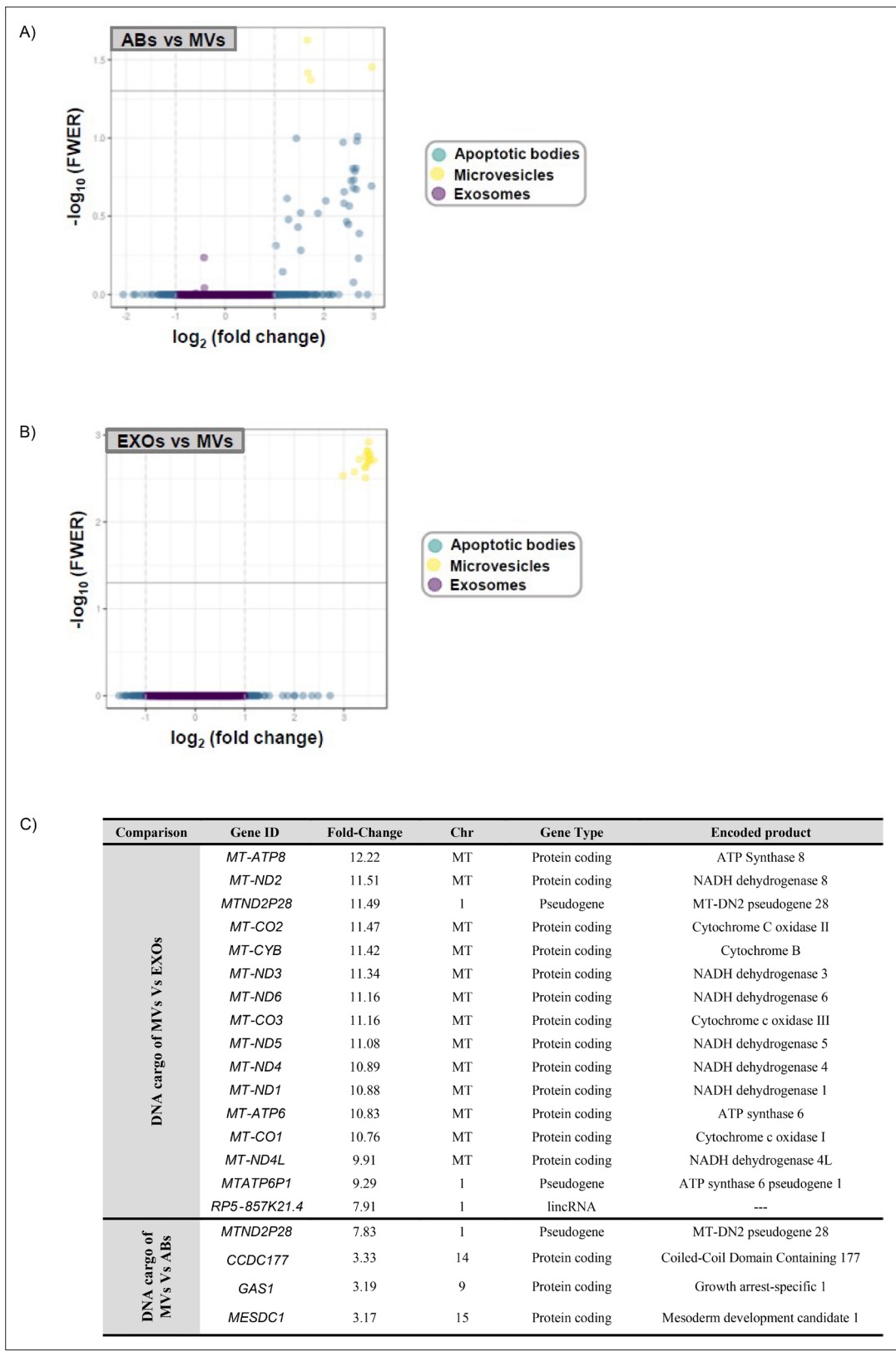

**Figure 2.** DNA sequencing analysis and coding sequence comparisons of human endometrial fluid-derived extracellular vesicle populations. (**A and B**) Volcano plots comparing DNA sequence enrichment between ABs, MVs, and EXOs. Only MVs show significant sequence enrichment compared to ABs and EXOs. (**C**) Specific gene ID DNA sequences encapsulated within MVs compared to ABs and EXOs, which are mainly mitochondrial DNA.

*Figure 2 continued on next page*

*Figure 2 continued*

The online version of this article includes the following figure supplement(s) for figure 2:

**Figure supplement 1.** Effect of DNase treatment in sequencing analysis and coding sequences comparison between human endometrial fluid-derived extracellular vesicle populations.

22 (*TOMM22*), when comparing the pre-receptive stage to the receptive and post-receptive stage (p<0.001; *Figure 3B*); these data may indicate a decrease in mitochondrial mass, which again coincides with a higher mtDNA content in secreted MVs.

As MVs possess a considerable content of mtDNA during the receptive phase, we measured the mtDNA copy number content in EF-derived MVs from patients undergoing HRT in the pre-receptive, receptive or periconceptional period, and post-receptive periods to compare against the findings from endometrial samples. Quantifying mtDNA copy number in MVs demonstrated the accumulation of mtDNA during the receptive compared to the pre-receptive period. Specifically, we found a 3.2-fold enrichment in mtDNA copy number in MVs in the receptive compared to the pre-receptive period and a 2.6-fold enrichment in the post-receptive compared to the pre-receptive period (*Figure 3C*); however, these changes failed to reach significance.

Altogether, these data indicate a reduction in mtDNA content and the activation of mitochondrial clearance mechanisms in the human endometrium at times associated with embryo implantation alongside the secretion of mtDNA in endometrial MVs.

## Vertical transmission of EV-derived DNA from the maternal endometrium to the embryo

To provide evidence for the internalization of EVs (and associated DNA) by murine embryos, we used the endometrial Ishikawa cell line to produce EVs containing DNA tagged with a synthetic molecule (Materials and methods) to support the tracking of EV-derived DNA uptake. Before confocal experiments, we first confirmed the ability of DNA to become internalized by EVs/packaged into EVs by flow cytometry (*Figure 4—figure supplement 1*). Confocal imaging demonstrated that ABs released from Ishikawa cells effectively transported DNA into the cells of the embryo, reaching the cytoplasm (*Figure 4*) and colocalizing with nuclei (Arrows in *Figure 4—figure supplement 2*). A considerable amount of AB-delivered DNA accumulated in large deposits in discrete zones of the embryo, but mainly in the cytoplasm (*Figure 4*). MVs delivered DNA into the trophectoderm of the embryo (*Figure 4* and *Figure 4—figure supplement 2*), with DNA transmission occurring in a widespread spotted pattern at the perinuclear level. We also observed evidence for the delivery of DNA into the embryo cell cytoplasm and nuclei by EXOs; of note, the small size of EVs makes their robust imaging more challenging than for ABs and MVs (*Figure 4* and *Figure 4—figure supplement 2*). Regardless of the subtype, EV-derived DNA internalized into hatched trophectoderm (sites where direct contact is possible; *Figure 4*). The control conditions (*Figure 4* - Neg), which contained cell-free DNA and residual small-sized EVs, failed to demonstrate any detectable signal indicating the transfer of DNA into the cells of the embryo, suggesting that DNA transport from the mother to the embryo at the endometrial level requires EVs (*Figure 4*). As final proof of the internalization of EV-derived DNA into the cytoplasm/nuclei of embryo cells, we constructed Z-stack/orthogonal projection images (*Figure 4—figure supplement 2*). To prove the internalization of mtDNA, we analyzed the incorporation of free (not encapsulated within EVs or other artificial vesicles) synthetic molecules of labeled mtDNA within embryos using confocal microscopy; overall, we also observed robust mtDNA internalization by trophectodermal cells of hatched embryos (*Figure 4—figure supplement 3*).

Overall, we demonstrate that DNA-containing EVs generated by human endometrial cells become internalized by the trophectoderm of murine embryos in vitro.

## Bioenergetic impact of endometrial EV-derived DNA uptake by the embryo

To assess the functional relevance of vertical endometrial EV-derived DNA transmission and the impact on embryo bioenergetics, we analyzed ATP concentrations in cells of the embryo. We co-cultured overnight hatched murine blastocysts with different human EF-derived EV populations (either separately or combined) derived from five donors during the receptive phase of the menstrual cycle

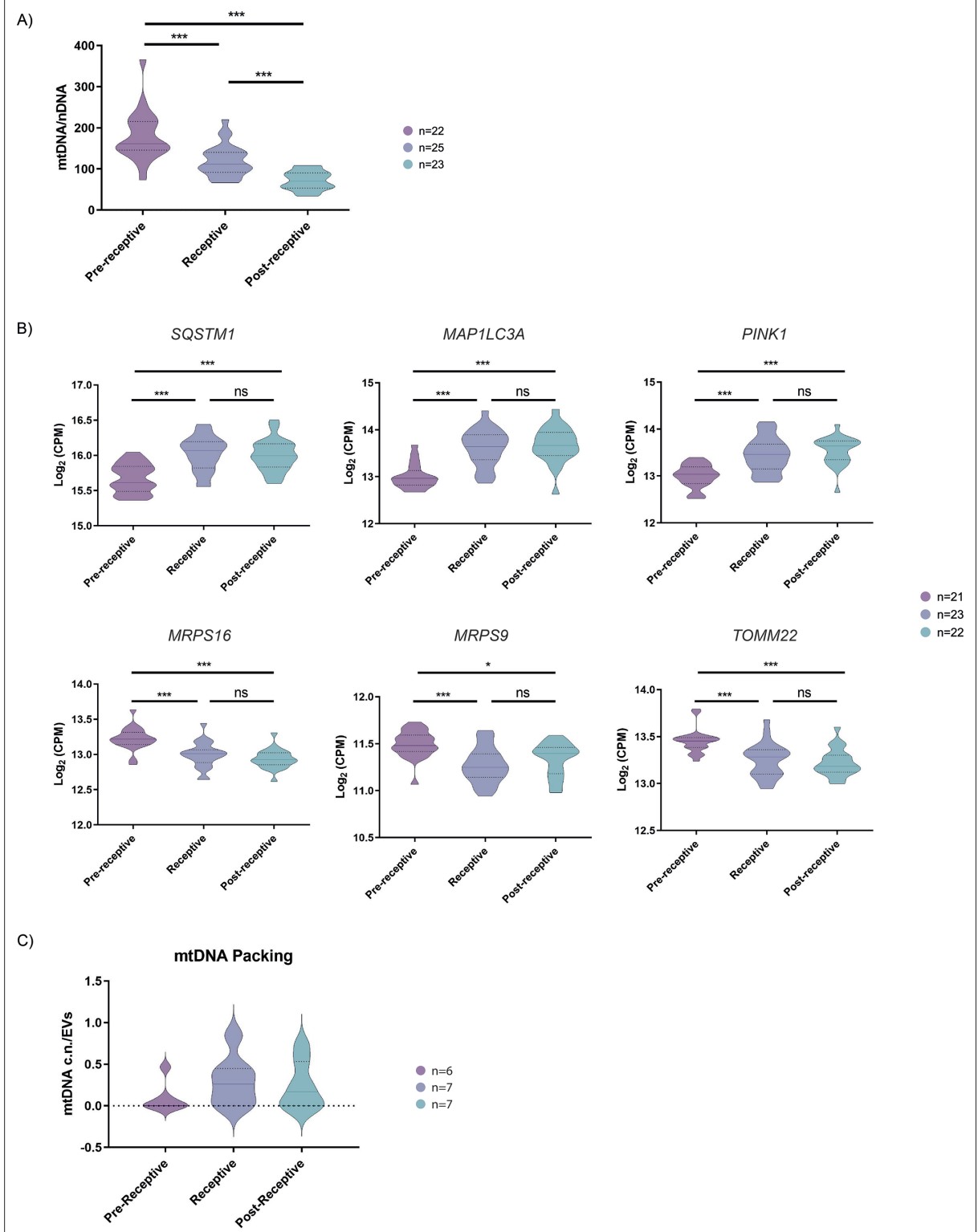

**Figure 3.** Quantification of mitochondrial DNA in human endometrial tissues and human endometrial fluid-derived microvesicles. (**A**) Relative mtDNA/nDNA ratio calculated from endometrial biopsies from donors undergoing HRT in pre-receptive (*P*+2), receptive (*P*+5), and post-receptive (*P*+8) periods, n=70. (**B**) Gene expression analysis of endometrial biopsies for nuclear genes coding for mitophagy- and mtDNA packing-related proteins (upper panel) and for genes coding for proteins related to mitochondrial function (lower panel), n=66. (**C**) Quantification of relative mtDNA copy number packed into MVs isolated from the EF in pre-receptive, receptive, and post-receptive periods, n=20. One-way ANOVA and Kruskal-Wallis rank sum tests performed to compare the distinct periods - no significant differences were observed. *p<0.05, ***p<0.001.

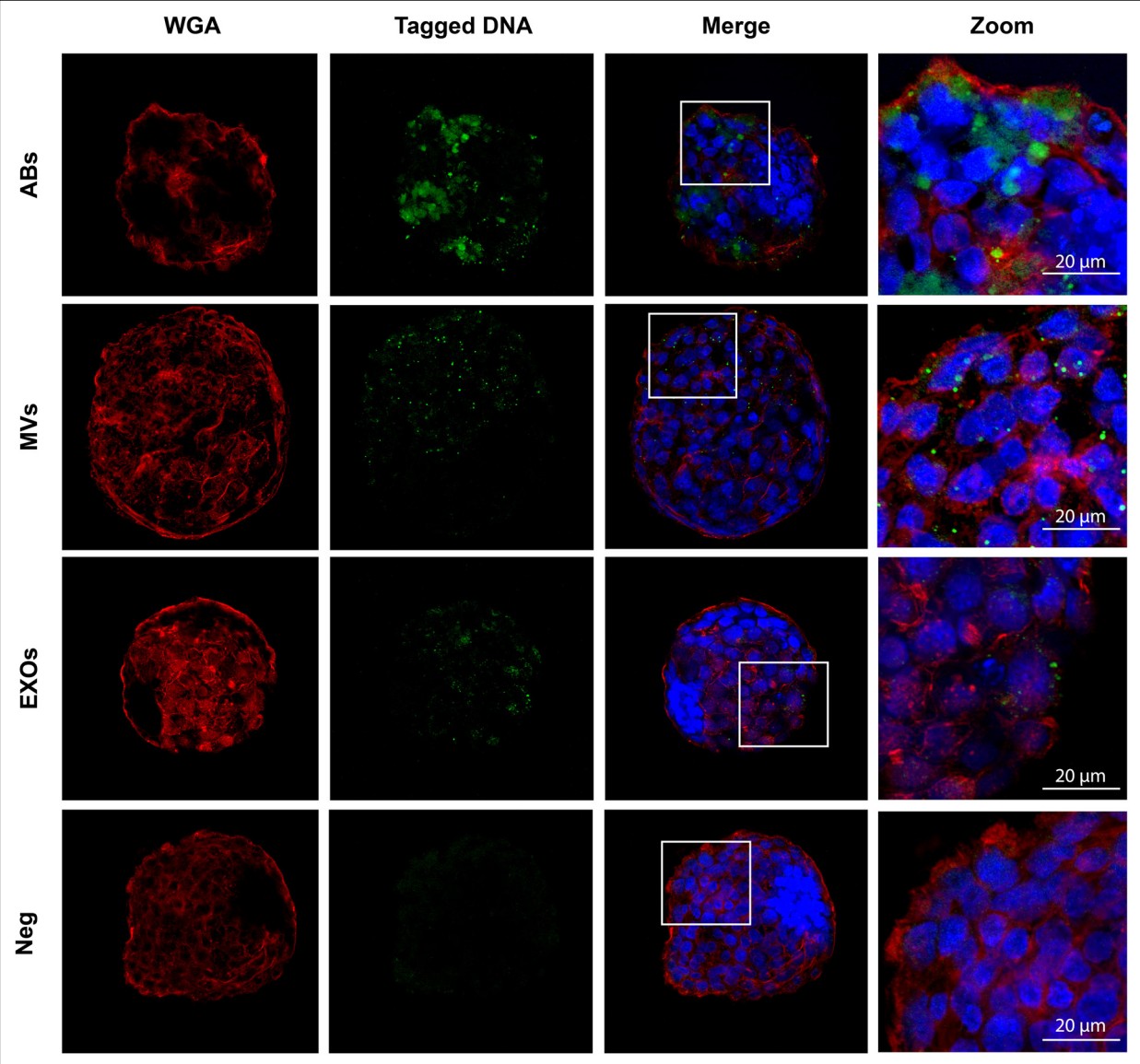

**Figure 4.** Internalization of endometrial extracellular vesicle-derived DNA by cells of the murine embryo. Confocal images show hatched embryos after co-culture with EdU-tagged ABs, MVs, and EXOs isolated from Ishikawa cell supernatants. Embryo membranes were visualized with Wheat germ agglutinin (WGA) in red, embryo nuclei with DAPI, and EdU-tagged transferred DNA in green. Zoomed images taken from the areas demarcated by white boxes in merge images. Cell-free DNA and residual small-sized EVs were used as control conditions (Neg). Scale bar in zoom = 20 µm.

The online version of this article includes the following figure supplement(s) for figure 4:

**Figure supplement 1.** Characterization of EdU-tagged DNA incorporation into extracellular vesicles isolated from Ishikawa cells.

**Figure supplement 2.** Z-stack/orthogonal sections of murine embryos co-cultured with extracellular vesicles containing EdU-tagged DNA.

**Figure supplement 3.** Detection of exogenous mitochondrial DNA in mouse embryos.

and compared results with those from embryos cultured in the absence of endometrial EVs (Control, Cnt). We found a significant reduction in ATP levels following embryo co-culture with 'All EVs' or EXOs compared to control embryos (both p<0.001; *Figure 5A*); however, murine embryos incubated with only ABs or MVs from human EF maintained ATP levels similar to control (p>0.05). These results suggest that EVs (EXOs in particular) significantly impact ATP consumption/production in the embryo. The observed reduction in ATP concentration after exposure to 'All EVs' and EXOs indicates an increase in cellular metabolic rate; therefore, embryos display ATP turnover after internalizing endometrial EVs, a situation that imitates the physiological state at periconception when the embryo comes into contact with the EF and EVs secreted by the endometrium.

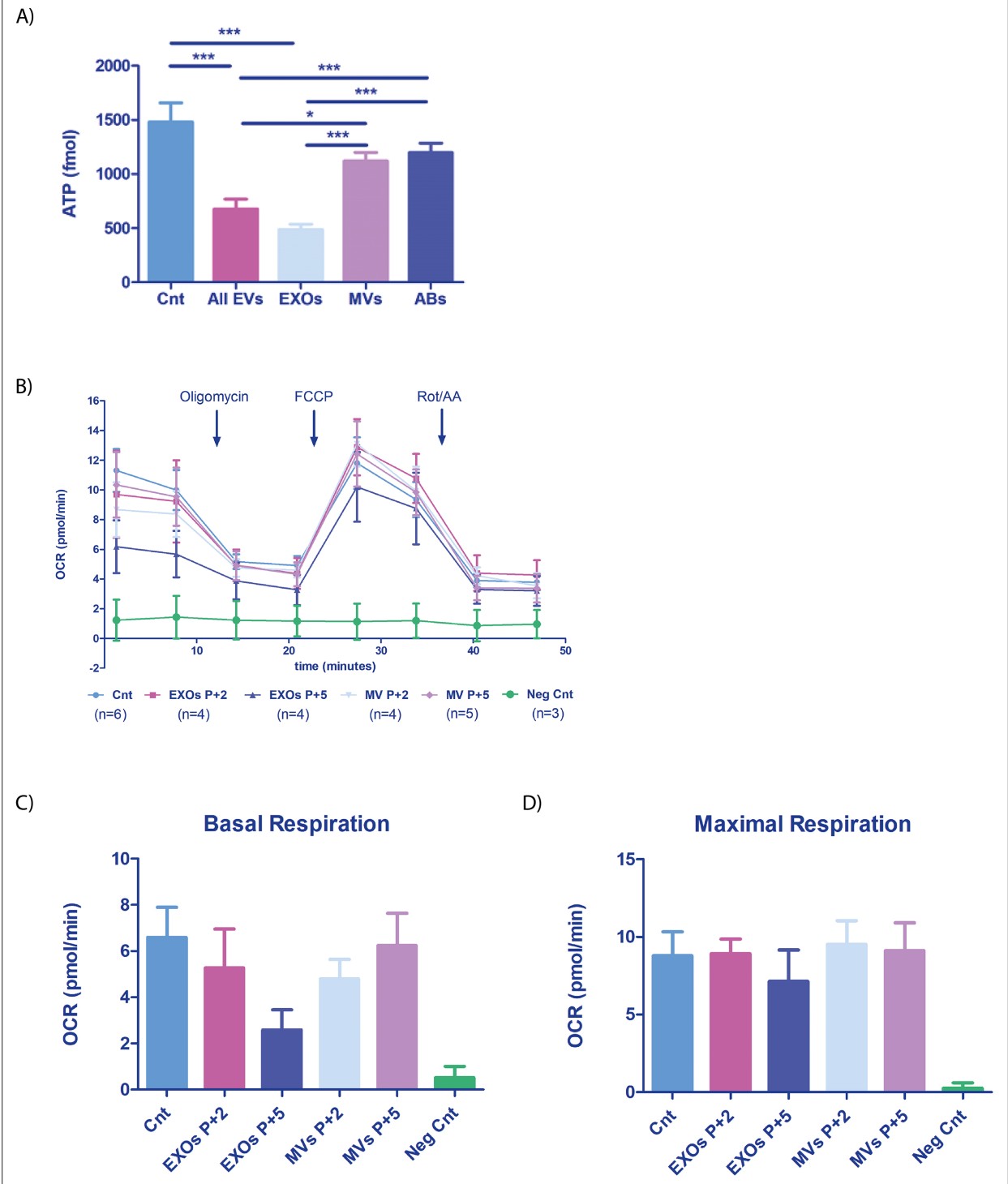

**Figure 5.** Mitochondrial function in embryos incubated with human endometrial fluid-derived extracellular vesicles. (**A**) Murine embryo ATP content (n=60) after overnight co-incubation with the EF-derived EV populations (phase IV or receptive phase of the natural menstrual cycle n=5). 'All EVs' indicates a combination of ABs, MVs, and EXOs. Embryos not incubated with EVs used as a control condition (Cnt). (**B**) OCR was recorded on a Seahorse instrument before and after drug injection (timing indicated on the graph). Blocked embryos used as an additional negative control (Neg Cnt), total embryos used (n=720). (**C**) Basal respiration [(Last rate measurement before the first injection)-(minimum rate measurement after Rotenone/antimycin A injection)] and (**D**) Maximal respiration [(Maximal rate measurement after FCCP injection)-(minimum rate measurement after Rotenone/antimycin A injection)] shown for each condition. One-way ANOVA and Tukey comparison post-hoc performed - no significant differences between conditions were observed (excluding the Neg Cnt condition). $*p < 0.05$, $***p < 0.001$.

To further understand the metabolic changes produced by EVs on the embryo, we studied the oxygen consumption rate (OCR) of embryos, as the manipulation of microenvironmental parameters to help uterine implantation represents a well-understood first signal from the blastocyst to the endometrium (*Gardner, 2015*; *Gardner and Harvey, 2015*; *Gurner et al., 2022*). We studied the effect of endometrial EVs by analyzing the OCR of embryos incubated with EF-derived MVs and EXOs (*Figure 5B*). We observed a reduction in the OCR of embryos treated with EXOs obtained from donors in the receptive phase (*P*+5) compared with the pre-receptive phase (*P*+2) (basal respiration and maximal respiration) (*Figure 5B–D*); while this reduction did not reach significance (perhaps due to the small sample size), the result paralleled the results observed for ATP consumption, potentially reflecting an increase in glycolytic flux. Of note, MVs appeared not to alter OCR (basal respiration and maximal respiration) in embryos (*Figure 5B–D*). Arrested embryos used as negative control (neg Cnt) did not respond to drug stimulation, indicating the viability of the embryos evaluated in this experiment.

These results suggest that EVs play a role in altering the bioenergetics of the embryo by altering the metabolic rate and oxygen consumption during periconception when a move towards increased glycolytic flux to support implantation is required. These results suggest that the endometrium may exchange signals with the embryo to aid the creation of conditions conducive to implantation.

## Discussion

We demonstrate the existence of endometrial EVs as potential vectors for the transport of DNA from the endometrium into the pre-implantation embryo; furthermore, our results suggest the vertical transmission of maternal DNA to the embryo as a mechanism to modulate embryo bioenergetics during the preconception period. These findings pave the way for more extensive, more detailed studies that aim to decipher the exact role of EV contents on embryo energetics, discover the relevance of packaged endometrial cell-derived DNA to this mechanism, and further expand our knowledge regarding materno-fetal communication.

We discovered that human endometrial cells secreted the three major EV subtypes into the EF during the receptive phase (phase VI) and characterized ABs, MVs, and EXOs morphologically using electron microscopy, molecularly using Western blotting, and by concentration and size distribution. In the case of MVs and EXOs, we identified similar concentrations in the EF throughout the menstrual cycle.

While other studies focused on the miRNA and protein content of EVs, we focused on the DNA cargo. We sequenced the DNA content of ABs, MVs, and EXOs isolated from the EF and found that all vesicles contained nDNA and mtDNA; however, MVs contained an ~elevenfold enrichment in 13 known mitochondrial genes (all coding for protein subunits that constitute the different complexes of the electron transport chain *Chinnery and Hudson, 2013*; *Taanman, 1999*). To understand the increased encapsulation of mtDNA within EVs, we quantified the mtDNA content of cells from endometrial biopsies, focusing on the possible differences between the pre-receptive, receptive, and post-receptive phases of the menstrual cycle. We observed a decrease in mtDNA content in endometrial cells during the receptive and post-receptive phases, which occurred at the same time as the upregulation of specific genes related to mitochondrial autophagy (*SQSTM1*, *MAPILC3A*, and *PINK1*), suggesting the existence of mitochondrial degradation in the endometrial tissue, coincident with the maximum encapsulation of mtDNA by MVs. *Holmgren et al., 1999* described a role for ABs in the horizontal transfer of DNA by phagocytosis, which the authors found to integrate into receiving cells and remain functional, as they can be rescued from the ABs and used by somatic cells. ABs support the formation of DNA molecular hybrids or hybrid chromosomes (*Bergsmedh et al., 2001*; *Waterhouse et al., 2011*) and transfer oncogenes that become internalized to increase target cell tumorigenic potential in vivo (*Bergsmedh et al., 2001*; *Ehnfors et al., 2009*). MVs also represent critical regulators of cancer pathogenesis (*Antonyak et al., 2011*; *Clancy et al., 2015*) in addition to their involvement in materno-fetal crosstalk (*Tong and Chamley, 2015*) and embryo self-regulation (*Desrochers et al., 2016*). Finally, EXOs play multiple roles in many biological processes; however, small RNA delivery (mainly miRNA) represents their primary function (*Valadi et al., 2007*; *Villarroya-Beltri et al., 2013*). The role of EVs in materno-fetal crosstalk and reproductive biology has been extensively analyzed (*Greening et al., 2016*; *Saadeldin et al., 2015*; *Vilella et al., 2015*). Interestingly, EXOs also play a role in packaging and transferring mtDNA from patients with hormonal therapy-resistant metastatic

breast cancer, which participates in the oncogenic signaling that induces cancer cells to exit dormancy (*Sansone et al., 2017*).

EVs containing DNA must first become internalized by trophectodermal cells to influence the embryo in any way. We demonstrated the internalization of ABs, MVs, and EXOs by murine embryos using labeled DNA tracked by confocal microscopy. In all experiments, we observed the internalization of EVs by the trophectodermal cells of murine embryos. ABs delivered substantial amounts of genetic material that occupied large but discrete regions of the embryo. In contrast, the MV-mediated DNA transmission pattern appeared widespread throughout the embryo as small perinuclear spots; meanwhile, EXOs also demonstrated potential for vertical DNA transfer.

Considering the DNA content of EVs and their uptake by the embryo, we evaluated a potential effect on embryo energetics. We hypothesized that maternal signals might regulate embryo bioenergetics to aid implantation (*Gardner, 2015*; *Gardner and Harvey, 2015*). Embryo co-culture with 'All EVs' (ABs, MVs, and EXOs) induced a lower ATP content than unstimulated controls, suggesting that the internalization of DNA (which includes the prevalent mtDNA content) may impact ATP consumption, which directly associates with the metabolic programming of embryos (*Gardner and Harvey, 2015*). We primarily observed the substantial reduction of ATP content when co-culturing murine embryos with isolated EXOs, suggesting that a combination of contents (e.g. DNA, RNA, proteins, and lipids) controls the metabolic state of embryos. Additional studies will aim to fully explore each EV cargo's potential contribution. We next studied the OCR of the embryos treated with EXOs and MVs from the pre-receptive and receptive phases (we removed ABs from this analysis due to their heterogeneity and methodological difficulties). We observed that EXOs obtained from receptive but not pre-receptive endometria appeared to reduce basal respiration (while not reaching significance), similar to the changes to ATP levels when treating embryos with 'All EVs' or EXOs. Again, future studies with more samples may reveal the real significance of these effects and decipher the exact role of each EV subtype and their cargos.

We suggest EVs as vectors for transporting nDNA and mtDNA into the pre-implantation embryo; this maternal DNA vertical transfer mechanism could influence embryo metabolism during the periconceptional period, suggesting that the mother sends signals to the embryo to aid the implantation process.

## Methods
### Experimental design
EF samples were collected from women undergoing natural (healthy volunteers aged 18–35) cycles and women undergoing hormonal replacement therapy. EF samples for the natural menstrual cycle were collected from each donor (one from each) and classified according to the menstrual cycle as phase I (early proliferative, days 0–8 (n=6)), phase II (late proliferative, days 9–14 (n=7)), phase III (early secretory, days 15–18; (n=7)), phase IV (mid-secretory or preconception period, days 19–24; n=36), and phase V (late secretory phase, days 25–30; (n=6)). Additional sets of EF samples were collected from women undergoing hormonal replacement therapy in pre-receptive (*P*+2), receptive (*P*+5), or post-receptive (*P*+8) stages (n=12 donors; three collections/donor) for a total of n=30 EF. The total number of EF samples collected for this study was n=62. Endometrial biopsies were collected from women undergoing hormonal replacement therapy in receptive (*P*+5) stages (n=70). The number of samples used in every experiment is specified throughout the methods section.

Inclusion criteria were patients with regular menstrual cycles, body mass index (BMI) of 18–30 kg/m$^2$, no contraceptive devices or hormonal treatment in the past three months, negative for bacterial/viral infectious diseases, and without obvious uterine or adnexal pathologies. The IRB committee approved this study at IVI Valencia, Spain (1603-IGX-017-FV and IGX1-VES-FV-21–04), and all subjects provided signed informed consent.

### Endometrial fluid sampling
After cleansing the vaginal channel, EF samples were obtained from the uterine fundus using a double lumen embryo transfer catheter (Wallace, Smiths Medical International, Minneapolis, MN, USA) introduced into the uterine cavity and applying gradual suction using a 10-mL syringe. The vacuum was

stopped to prevent contamination with cervical mucus, the catheter was gently removed, and EF was introduced in an Eppendorf tube for storage at –80 °C until processing.

## EV isolation from endometrial fluid

Each EF sample was resuspended in 1 mL of cold (4 °C) Dulbecco's PBS without $Ca^{2+}$/$Mg^{2+}$ (L0615-500; Biowest, Barcelona, Spain) to prevent salt precipitation. Resuspensions were homogenized by extensive pipetting and vortex, and samples were treated with 50 U/mL DNase type I (D4513; Sigma-Aldrich, Madrid, Spain) to disaggregate mucus and eliminate extravesical DNA. For EV isolation, each resuspended EF sample volume was increased to 4 mL with PBS, and samples underwent a series of differential centrifugations and filtration; centrifuged twice at 300 x $g$ for 10 min to pellet residual cells and debris, with the resulting supernatants centrifuged at 2000 x $g$ for 10 min, passed through 0.8 µm-diameter filters (GE Healthcare, Life Sciences, Whatman, UK), centrifuged at 10,000 x $g$ for 30 min, passed through 0.22 µm-diameter filters (Acrodisc syringe filters, Pall Corp., Newquay, Cornwall, UK), and ultracentrifuged at 120,000 x $g$ for 70 min using a P50AT4 rotor (Hitachi Koki Co. Ltd., Tokyo, Japan). Pellets were washed in 1 mL PBS and centrifuged again under the same conditions to obtain fractions subsequently enriched in ABs (2,000 x $g$ for 15 min), MVs (10,000 x $g$ for 40 min), and EXOs (120,000 x $g$ for 70 min, using a Hitachi P50A3 rotor). The resulting supernatants were kept as negative controls. All centrifugations were carried out at 4 °C.

## Transmission electron microscopy

Pellets from serial differential centrifugations of single EF samples (n=4, 2 per each technique) obtained in phase IV of the natural cycle were analyzed by two TEM techniques.

*Deposition and positive staining* (to evaluate external aspects and preliminary size): Pellets were resuspended in 50 µL of Karnovsky's fixative solution (2.5% glutaraldehyde/2% formaldehyde in 0.1 M phosphate buffer, pH = 7.4), and a drop of the resulting mix was laid onto a 300-mesh grid. Samples were then incubated with 2% uranyl acetate in Reynold's lead citrate solution (80 mM lead nitrate, 120 mM sodium citrate dihydrate, and 100 mM NaOH in distilled water).

*Inclusion in LR-white resin and ultrathin cuts* (to evaluate EV internal structures and general contents): Isolated pellets were carefully fixed in 50 µL of Karnovsky's solution without disturbing the pellet. Fixed pellets were washed five times in PBS for 5 min each and stained in a 2% osmium tetroxide/0.2 M PBS solution for 2 hr. Samples were dehydrated, embedded in resin (Epoxy), and ultra-sectioned in 60 nm slices. Samples were observed using a JEM-1010 TEM (Jeol Korea Ltd., Seoul, South Korea) at 80 kV coupled to a digital camera MegaView III.

## Western blotting

EV populations isolated from phase IV of natural cycle human EF (n=3) were lysed in 50 µL of RIPA buffer (150 mM NaCl, 1% IGEPAL CA 630, 0.5% Na-DOC, 0.1% SDS, 0.5 M EDTA, 50 mM Tris-HCl, pH 8) prepared with protease inhibitors [89% RIPA, 1% 0.1 M PMSF (Sigma-Aldrich, Madrid, Spain), 10% Roche Mini Complete (Roche, Madrid, Spain)]. The protein content of EV samples, supernatants obtained during EV isolation, and tissue cell lysates were quantified by Bio-Rad Protein Assay (Bio-Rad Laboratories, Hercules, CA, USA). The cell lysate was generated as a Western blot positive control from the human endometrial epithelium Ishikawa cell line (Sigma-Aldrich, Madrid, Spain), as in a previous study (*Vilella et al., 2015*). Equal protein amounts were denatured at 95 °C, electrophoresed by SDS-PAGE, and transferred to PVDF membranes (Bio-Rad Laboratories).

Membranes were blocked in 5% skim milk in 1% Tween-20 in PBS (PBS-T) and incubated overnight at 4 °C with the following primary antibodies: rabbit anti-calnexin (1:1000; ADI-SPA-865; Enzo Life Sciences, Farmingdale, NY, USA), mouse anti-calreticulin (1:1000; ab22683; Abcam, Cambridge, UK), rabbit anti-VDAC1 (1:1000; ab154856; Abcam), rabbit anti-ARF6 (1:1000; ab77581; Abcam), mouse anti-CD63 (1:1000; ab59479; Abcam), rabbit anti-CD9 (1:1000; ab92726; Abcam), and rabbit anti-TSG101 (1:1000; 125011; Abcam). The following day, membranes were washed and incubated with secondary antibodies: goat anti-mouse (1:10,000; sc-2005; Santa Cruz Biotechnology, Dallas, TX, USA) or goat anti-rabbit (1:20,000; EXOAB-KIT-1; System Biosciences, Palo Alto, CA, USA). Protein signals were detected using the SuperSignal West Femto Chemiluminescent kit (Thermo Fisher Scientific, Waltham, MA, USA) and a LAS-3000 imaging system (Fujifilm, Japan).

## Dynamic light scattering

Two EF samples were used for the ZetaSizer Nano (Malvern Instruments Corp., Malvern, UK), a device based on dynamic light scattering (DLS) technology, which was used to generate general size distribution patterns of the different EV fractions in a wide size range (1–6000 nm). For analysis, pellets from serial differential centrifugation steps were resuspended in 1 mL of PBS without $Ca^{2+}/Mg^{2+}$, transferred to 4 mL polystyrene cuvettes, and analyzed on a Malvern ZetaSizer Nano-ZS 90.

## Nanoparticle tracking analysis

EF samples for the natural menstrual cycle, phase I (early proliferative, days 0–8 [n=6]), phase II (late proliferative, days 9–14 [n=7]), phase III (early secretory, days 15–18; [n=7]), phase IV (mid-secretory or preconception period, days 19–24; n=6), and phase V (late secretory phase, days 25–30; (n=6)). NanoSight 300 (Malvern Instruments Corp.), a technology based on nanoparticle tracking analysis (NTA) principles, was used to finely analyze EV size distribution and concentrations from EF samples throughout the menstrual cycle using light scattering properties. Due to its more limited size working range (up to 1000 nm), ABs could not be analyzed. To normalize concentration measurements, EF volumes were measured prior to EV isolation so that we could refer to EV concentration per mL. Pellets containing isolated vesicles were resuspended in 1 mL PBS without $Ca^{2+}/Mg^{2+}$ and introduced into the NanoSight 300.

## High throughput sequencing of the DNA cargo from human endometrial EVs

To analyze the DNA cargo of the different EV populations within the human EF, EVs were isolated from single EF samples in phase IV (n=10) of the menstrual cycle as described above. To eliminate DNA stuck externally to the vesicles, isolated EVs were treated with 50 U/mL DNaseI (Sigma-Aldrich, ref D4513) in a solution containing 20 mM Tris-HCl, 10 mM $MgCl_2$ and 1 mM $CaCl_2$ and incubated at 37 °C for 30 min while gently shaking. DNase type I pre-treatment was performed to ensure that bona fide DNA cargo was sequenced. To prove the use of DNase, a total of six EF phase IV samples were pooled and separated in fractions and divided into three different groups: control group (no DNase), treated with DNaseI, and treated with DNaseI +exonuclease T5; the PCA for coding gene sequences demonstrated that DNase treatment accounted for 56% and 57% of the total variability for ABs and EXOs, respectively, in PC1 (*Figure 2—figure supplement 1A, B*). Associated volcano plots revealed abundant enrichment in coding sequences over whole genomic DNA after DNase treatment (Yellow dots). These results indicated that external DNA contamination masks EVs DNA cargo, which should be prevented by DNase treatment to investigate specific EVs DNA inner cargo.

After digestion, DNaseI was heat-inactivated by incubating at 75 °C for 10 min at 2:30 v:v of 0.5 M EDTA (Thermo Fisher Scientific, ref 15575020). Next, DNA contained in EVs was extracted using the QIAamp DNA Mini Kit protocol for cultured cells (Qiagen, ref 51306). Only samples that accomplished a minimum of 0.03 ng/μL of DNA in all the vesicle populations from the same EF were chosen for library construction. A lower threshold was chosen based on our previous experience. Samples over 0.2 ng/μL were adjusted to this concentration with nuclease-free water. To analyze the effect of lowering the DNA input amount for libraries construction (samples with initial limiting input DNA), those samples with DNA excess from the DNase evaluation assay (i.e. DNase untreated EXOs and ABs) were used to create serial dilutions for library construction. The resulting libraries containing an initial 0.05 ng/μL DNA and those in the ideal concentration range (0.2 ng/μL and 0.15 ng/μL for ABs and EXOs, respectively) were chosen to evaluate the effect of reducing DNA input for sequencing. The selection was based on the similarity of DNA concentrations and library profiles, measured by Bioanalyzer 2100 HS DNA assay (Affymetrix), between 0.05 ng/μL DNA dilutions and initial limiting DNA samples.

In all cases, libraries were constructed using the Nextera XT DNA Library Prep Kit (Illumina, ref FC-131–1024) following the manufacturer's instructions. As variable parameters, libraries AMPure XP (Beckman Coulter, ref 082A63881) purification was carried out at 1 X, bead-based normalization was chosen, and, following Illumina's instructions, a 4 μL volume was chosen for serial dilution in 996 μL and 750 μL/750 μL in HT1. Finally, libraries were sequenced by an Illumina NextSeq 500 (Illumina) using a 300-cycle NextSeq 500 High Output v2 Kit cartridge (Illumina, ref FC-404–2004).

## DNA sequencing analysis

Raw data from pair-ended Illumina sequencing was converted into FASTQ files using bcl2fastq (version 2.16.0.10). Raw counts were normalized using the TMM method from the edgeR R package, and each sample was aligned to the reference genome (GRCh37) using BWA (version 0.7.10). Reads with mapping quality >90% were filtered using Samtools (version 1.1), and duplicates were removed with PICARD software. Insert size was retrieved from filtered reads using PICARD software, and feature coverage was obtained with Bedtools (version 2.17.0) using Ensembl Biomart hg19 annotations. For the following bioinformatics analysis, reads mapping to chromosome Y and noisy samples were filtered out. The approach used for differential DNA enrichment analysis was based on the edgeR methodology. PCA graphs were obtained from log2 normalized CPM using the prcomp R function for all samples and comparisons. Descriptive and Pearson's correlation analyses were carried out to analyze the effect of reducing initial DNA concentration.

## Endometrial biopsies

All patients underwent an HRT cycle for endometrial preparation (n=70). Endometrial biopsies were collected from the uterine fundus using a pipette under sterile conditions. The day of the endometrial biopsy is calculated after estrogen priming, leading to a trilaminar endometrium measuring ≥6.5 mm after five full days of progesterone impregnation (~120 hr). After the biopsy, the endometrial tissue was transferred to a cryotube containing 1.5 mL RNAlater (Qiagen), vigorously shaken for a few seconds, and kept at 4 °C or on ice for ≥4 hr. Determining the receptivity window used the ERA test, as previously described (*DiazGimeno et al., 2011*). Biopsies were classified as receptive (n=25), pre-receptive (n=22), and post-receptive (n=23).

## Determination of mitochondrial DNA copy number in biopsies

Relative amounts of nDNA and mtDNA were determined by quantitative RT-PCR as previously described (*DiezJuan et al., 2015*). The nuclear gene β-actin was selected as a housekeeping gene for normalization. mtDNA was quantified using an mtDNA fragment within the ATP8 gene. The ratio of mtDNA/nDNA was used to indicate the mitochondrial copy number per cell.

## Quantitative RNA sequencing in Biopsies

A total of 100 ng total RNA was reverse transcribed using the Ion AmpliSeq Transcriptome Human Gene Expression Kit following the manufacturer's protocol (Thermo Fisher Scientific). Target genes (*SQSTM1, MAP1LC3A, PINK1, MRPS16, MRPS9, and TOMM22*) were amplified using an Ion AmpliSeq Transcriptome Human Gene Expression Core Panel (Thermo Fisher Scientific). Amplicons were ligated to barcode adapters and purified using Agencourt AMPure XP reagent (Beckman Coulter Inc). After purification, amplicons were eluted and normalized before emulsion PCR and chip loading on the Ion Chef system, obtaining reads with an average length of 200 bp. The sequencing data were aligned using the ampliSeqRNA plugin for Ion Torrent sequencing platforms (Thermo Fisher Scientific).

## mtDNA copy number quantification in MVs

As detailed above, EVs isolated from EF (n=21) in different menstrual cycle stages (*P*+2 n=7, *P*+5 n=7, *P*+8 n=7) were DNase treated to eliminate extravesical DNA contamination. MVs-derived DNA was amplified using the Sureplex DNA amplification system (Illumina) according to the manufacturer's instructions. This amplification step increases the DNA yield obtained per sample to ensure a sufficient amount for qPCR analysis without altering the proportion of the different sequences. Samples were purified using AMPure XP beads (Beckman Coulter), and the DNA was quantified using D1000 Scre-enTape in a TapeStation 4200 instrument (Agilent). Only DNA samples containing >0.1 ng/μL were used for the qPCR analysis. The mitochondrial ATP8 gene was amplified in a qPCR machine using the following primers: Forward 5'-CTAAAAATATTAAACACAAACTACCACCTACCTC-3' and Reverse 5'-GTTCATTTTGGTTCTCAGGGTTTGTTATAA-3'. Standard curves were created by qPCR amplifying serial dilutions of the ATP8 amplicon and used to calculate the copy number of mtDNA in 250 pg of DNA per reaction. Half of the MVs samples were used to calculate the concentration of MVs by NTA analysis, and the mtDNA copy number was normalized with the number of MVs (mtDNA copy number/MV).

## Murine embryo isolation and culture

Murine embryos were obtained from B6C3F1/Crl mice (Charles River Laboratories, Saint-Germain-Nuelles, France). The Animal Care and Use Committee of Valencia University (CEBA) authorized the project under the identifier: 2015/VSC/PEA/00048. The embryo recovery and culture processes were adapted from our previous work (*Vilella et al., 2015*). Briefly, female mice aged 6–8 weeks were stimulated to ovulate by 10 IU intraperitoneal injection of Foligon/PMSG (MSD Animal Health, Spain) followed by intraperitoneal administration of 10 IU of Ovitrelle 250 µg/0.5 mL (Merck Serono, Germany) 48 hr later. At that point, females were mated with males of the same strain for 48 hr, checking for the presence of a vaginal plug. Plug-positive females were sacrificed by cervical dislocation, and embryos were collected from the oviduct by flushing with PBS using a 30-gauge blunt needle. Embryos were then washed four times in overnight oxygenated G-2 plus medium (G-2 PLUS, Vitrolife, Barcelona, Spain) and cultured until hatching for 48 hr in the same medium (day E3.5 of embryo development). An average of 30–40 embryos per female were obtained, with 60% reaching the hatching state with excellent quality at E3.5.

## Tagged-DNA production and EV internalization by murine embryos

The human authenticated Ishikawa cell line were bought on Sigma Aldridge (99040201-DNA-5UG), that came from ECACC, and tested as mycoplasma negative prior to perform the experiment. The cells were grown in flasks in Modified Eagle's Medium (MEM, Gibco, Thermo Fisher Scientific, ref 10370021), supplemented with 5% fetal bovine serum (Biowest, Barcelona, Spain, ref S181B-500), 1% non-essential amino acids (Gibco, Thermo Fisher Scientific, ref 11140035), 1% Glutamax (Gibco, Thermo Fisher Scientific, ref 35050–038), 0.2% 50 mg/mL gentamicin (Gibco, Thermo Fisher Scientific, ref 15750–037) and 0.2% 250 µg/mL amphotericin B (Gibco, Thermo Fisher Scientific, ref 15290026), until 60–70% confluence.

For the generation of EdU DNA-tagged EVs, three T175 flasks of Ishikawa cells at 60–70% confluence were supplemented with 1 µM 5-ethynyl-2'-deoxyuridine (EdU, Thermo Fisher Scientific) and incubated overnight to allow EdU incorporation into the DNA. The next day, cells were washed twice in PBS and added to the Ishikawa medium containing EV-depleted fetal bovine serum. Finally, conditioned media containing EVs was collected after 24 hr, and the different EVs populations were isolated as described above.

Pellets containing Ishikawa EdU DNA-tagged EVs were added to good-quality E3.5 hatching embryos and co-incubated overnight. Twenty embryos were used for each condition (ABs, MVs, EXOs, and Neg), for a total of eighty embryos. The supernatant of the isolation process and a mixture of EVs populations generated by Ishikawa cells cultured in the absence of EdU were used as negative controls. After embryo co-culture with EVs, transferred EV-DNA was stained by click-it chemistry reaction using Click-iT EdU Alexa Fluor 488 Imaging Kit (Thermo Fisher Scientific, ref: C10337). The protocol was carried out as recommended by the manufacturers with some modifications. Embryos were fixed with 3.7% formaldehyde in PBS solution and stained with Wheat Germ Agglutinin, Texas Red-X Conjugate (Thermo Fisher Scientific, ref: W21405) for 20 min at 37 °C. After permeabilization in 0.5% Triton X-100 in PBS, embryos were labeled with EdU, rinsed in PBS, and stained with DAPI solution (Thermo Fisher Scientific, ref: 62248). Stained embryos were imaged with an FV1000 Olympus confocal microscope using a 60 X oil-immersion lens.

The Click-iT EdU Alexa Fluor 488 flow cytometry assay Kit (Thermo Fisher Scientific, ref: C10420) was used to analyze EdU incorporation within the different EV subpopulations obtained from Ishikawa cells. A fraction of the isolated EVs from EdU-labeled and non-labeled Ishikawa cells were pelleted and resuspended with the Click-it Plus reaction cocktail following the manufacturer's instructions. After a washing step, EVs were analyzed in a cytoFLEX flow cytometer (Beckman Coulter), and the data were analyzed with FlowJo software.

## mtDNA internalization by mouse embryos

Hatched mouse embryos (3.5 days) were co-culture with an ATP8 sequence (Forward: /5Biosg/ctaa aaatattaaacacaaactaccacctacctccctcaccaaagcccataaaaataaaaaattataacaaaccctgagaaccaaaatgaac; Reverse: /5Biosg/gttcattttggttctcagggtttgttataatttttttatttttatgggctttggtgagggaggtaggtggtagtttgtgtt taatattttag) conjugated with Biotin (manufactured by IDT) overnight at 37 °C and 5% $CO_2$. Embryos were permeabilized with 0.1% Triton-X-100 in PBS for 20 min, blocked with 3% bovine serum albumin

in PBS for 30 min shaking, incubated with Streptavidine-Cy3 (S6402, Merck; dilution 1:100) for 45 min, washed twice in PBS, and then visualized using an SP8 confocal microscope (Leica).

## Analysis of embryo ATP level modulation after co-incubation with human EVs

To analyze the impact of EVs in embryo bioenergetics modulation, EF samples (n=5) in phase IV of the menstrual cycle were pooled, and their EV populations were isolated as described above. Embryos from ten mice were obtained and incubated until day E3.5, when only hatching embryos were co-incubated with isolated EF-derived EVs overnight. The following day, embryos were washed in PBS, collected in 1 µL PBS, and passed into 96-well opaque plates (Sigma-Aldrich, ref: CLS3917-100EA) in 50 µL $H_2O$. Replicates of ten good-quality embryos were employed, achieving six replicates (n=60) for ABs and MVs, five for EXOs and all EVs conditions, and three for fresh G2 condition media. One µL PBS was included in duplicate in 50 µL $H_2O$ as a control of the embryo transport vehicle, and an ATP standard curve was generated with fifteen total ATP points from 5 fmol to 50 pmol in duplicate (including a blank) in 50 µL $H_2O$. ATP measurements were carried out using an ATP bioluminescent somatic cell assay kit (FLASC, Sigma), adapting the protocol from previous studies of human/murine blastomeres and oocytes (*Van Blerkom et al., 1995*; *Stojkovic et al., 2001*). Stock solutions were prepared following the manufacturer's instructions. The reaction mix was prepared by diluting FLAAM working solution 1:10 in FLAAB and incubating for 5 min at 28 °C in the dark. Samples on the plate were added 100 µL 1 X ice-cold FLSAR, incubated for 5' at 4 °C, and continued processing in a Clario-Star BMG Labtech (BMG Labtech, Germany, Software Version 5.21 .R2). The device dispensed 100 µL reaction mix and measured luminescence produced by the luciferase reporter system in consecutive cycles of 1.8 s reaction mix injection, 2 s shaking, and 2 s measurement. The gain was adjusted with the higher ATP standard.

## Mitochondrial function assays in embryos

The Seahorse XFe96 Extracellular Flux Analyzer instrument (Agilent) was used to measure the OCR of cultured murine embryos. Hatching/hatched stage embryos were incubated with pre-receptive ($P$+2; n=4) and receptive stage ($P$+5; n=5) EF-derived EVs $P$+2 derived EXOs (n=4; total embryos: 110); $P$+2 derived MVs (n=4; total embryos: 130); $P$+5 derived EXOs (n=4; total embryos: 120); $P$+5 derived MVs (n=5; total embryos: 150) overnight. Embryos incubated without EVs were used as a control group (n=6; total embryos: 160), while blocked embryos were used as negative control (n=3; total embryos: 120). A total of 720 murine embryos were used for this experiment. Embryos were seeded at ten embryos/well with XF DMEM medium containing 10 mM glucose, 2 mM L-glutamine, and 1 mM pyruvate and stimulated sequentially with 1 µM oligomycin, 1.5 µM carbonyl cyanide-p-trifluoromethoxyphenylhydrazone (FCCP), and 0.5 µM rotenone +antimycin A. Three/four replicates per condition were used in each experiment.

## Data analysis

ZetaSizer Nano corresponding to particle concentrations for different EV populations were obtained as an average value from five independent measures of random NTA fields. The standard error for different measures was calculated and represented for each curve as an indicator of the evenness of particles across the sample.

To evaluate variation in vesicle concentration throughout the menstrual cycle, concentration data for different samples were uploaded into R software, and the Kruskal-Wallis algorithm was used to detect differences.

ANOVA test, Statistical analysis, and Tukey multiple pairwise-comparisons test were used for conditions comparison in ATP content of murine embryos, mitochondrial respiration assays in Seahorse instrument experiments, and mtDNA copy number in EVs. Normal distribution in the study groups was assumed, and p-values <0.05 were considered significant.

## Acknowledgements

This work was supported by FIS projects from the Spanish Instituto de Salud Carlos III (ISCIII) [PI18/00957 and PI21/00528] to FV and [PI21/00235] to IM. DB was supported by a Formación de Personal Universitario grant [FPU15/02248] by the Spanish Ministerio de Educación, Cultura

y Deporte. JGF was supported by Contratos Predoctorales de Formación en Investigación en Salud (PFIS) from ISCIII [FI19/00159]. FV was supported by Miguel Servet Program Type II of ISCIII [CPII18/00020].

## Additional information

### Funding

| Funder | Grant reference number | Author |
| --- | --- | --- |
| Instituto de Salud Carlos III | PI18/00957 | Felipe Vilella |
| Instituto de Salud Carlos III | PI21/00528 | Felipe Vilella |
| Instituto de Salud Carlos III | PI21/00235 | Inmaculada Moreno |
| Ministerio de Educación, Cultura y Deporte | FPU15/02248 | David Bolumar |
| Instituto de Salud Carlos III | FI19/00159 | Javier Gonzalez-Fernandez |
| Instituto de Salud Carlos III | CPII18/00020 | Felipe Vilella |

The funders had no role in study design, data collection and interpretation, or the decision to submit the work for publication.

### Author contributions

David Bolumar, Conceptualization, Formal analysis, Investigation, Methodology, Writing - original draft; Javier Moncayo-Arlandi, Formal analysis, Investigation, Methodology, Writing - original draft; Javier Gonzalez-Fernandez, Ana Monteagudo-Sanchez, Carlos Marin, Investigation, Methodology, Writing – review and editing; Ana Ochando, Methodology, Writing – review and editing; Inmaculada Moreno, Antonio Diez, David K Gardner, Formal analysis, Writing – review and editing; Paula Fabra, Writing – review and editing, obtained samples; Miguel Angel Checa, Writing – review and editing, obtained samples; Juan Jose Espinos, Writing – review and editing, obtained samples; Carlos Simon, Conceptualization, Resources, Supervision, Funding acquisition, Validation, Writing - original draft, Writing – review and editing; Felipe Vilella, Conceptualization, Resources, Formal analysis, Supervision, Funding acquisition, Validation, Investigation, Methodology, Writing - original draft, Writing – review and editing

### Author ORCIDs

David Bolumar http://orcid.org/0000-0002-6234-3821
David K Gardner http://orcid.org/0000-0003-3138-8274
Felipe Vilella http://orcid.org/0000-0002-0039-9846

### Ethics

The IRB committee approved this study at IVI Valencia, Spain (1603-IGX-017-FV and IGX1-VES-FV-21-04), and all subjects provided signed informed consent.
Murine embryos were obtained from B6C3F1/Crl mice (Charles River Laboratories, Saint-Germain-Nuelles, France). The Animal Care and Use Committee of Valencia University (CEBA) authorized the project under the identifier: 2015/VSC/PEA/00048.

Reviewer #1 (Public Review): https://doi.org/10.7554/eLife.88008.4.sa1
Reviewer #2 (Public Review): https://doi.org/10.7554/eLife.88008.4.sa2
Author Response https://doi.org/10.7554/eLife.88008.4.sa3

## Additional files

### Supplementary files
• MDAR checklist

## Data availability

All data from this manuscript is included in the manuscript.

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
