## [Editor Report · eLife assessment]

This manuscript reports **important** results on the potential influence of maternally derived extracellular vesicles on embryo metabolism. The study combines **convincing** techniques for isolating different subtypes of EV, DNA sequencing, embryo culture, and respiration assays performed on human endometrial samples and mouse embryos. These findings set the stage for in-depth studies to elucidate the role of EV contents in embryo energetics and further enhance our understanding on maternal-fetal communication during peri-implantation development.

---

## [Referee Report · Reviewer #1 (Public Review)]

Bolumar et al. isolated and characterized EV subpopulations, apoptotic bodies (AB), Microvesicles (MV), and Exosomes (EXO), from endometrial fluid through the female menstrual cycle. By performing DNA sequencing, they found the MVs contain more specific DNA sequences than other EVs, and specifically, more mtDNA were encapsulated in MVs. They also found a reduction of mtDNA content in the human endometrium at the receptive and post-receptive period that is associated with an increase in mitophagy activity in the cells, and a higher mtDNA content in the secreted MVs was found at the same time. Last, they demonstrated that the endometrial Ishikawa cell-derived EVs could be taken by the mouse embryos and resulted in altered embryo metabolism.

This is a very interesting study and is the first one demonstrating the direct transmission of maternal mtDNA to embryos through EVs.

---

## [Referee Report · Reviewer #2 (Public Review)]

In Bolumar, Moncayo-Arlandi et al. the authors explore whether endometrium-derived extracellular vesicles contribute DNA to embryos and therefore influence embryo metabolism and respiration. The manuscript combines techniques for isolating different populations of extracellular vesicles, DNA sequencing, embryo culture, and respiration assays performed on human endometrial samples and mouse embryos.

Vesicle isolation is technically difficult and therefore collection from human samples is commendable. Also, the influence of maternally derived DNA on the bioenergetics of embryos is unknown and therefore novel.

---

## [Author Response]

The following is the authors’ response to the previous reviews

eLife assessmentThe manuscript offers important findings on the potential influence of maternally derived extracellular vesicles on embryo metabolism. However, while the content is convincing, the title appears to overstate the study's conclusions due to its speculative nature on the DNA transmission and embryo bioenergetics connection. A more measured title would better represent the evidence presented.

We want to extend our heartfelt appreciation to the editors and reviewers for their invaluable comments on our research. Their feedback has played a crucial role in improving the quality of our manuscript.

We acknowledge the concern regarding the manuscript's title and are fully open to making modifications. Following the recommendation of Reviewer 2, the proposed new title of the manuscript will be “Vertical transmission of maternal DNA through extracellular vesicles associates with altered embryo bioenergetics during the periconception period.”

**Reviewer #1 (Public Review):**
Q1. Bolumar et al. isolated and characterized EV subpopulations, apoptotic bodies (AB), Microvesicles (MV), and Exosomes (EXO), from endometrial fluid through the female menstrual cycle. By performing DNA sequencing, they found the MVs contain more specific DNA sequences than other EVs, and specifically, more mtDNA were encapsulated in MVs. They also found a reduction of mtDNA content in the human endometrium at the receptive and post-receptive period that is associated with an increase in mitophagy activity in the cells, and a higher mtDNA content in the secreted MVs was found at the same time. Last, they demonstrated that the endometrial Ishikawa cell-derived EVs could be taken by the mouse embryos and resulted in altered embryo metabolism.This is a very interesting study and is the first one demonstrating the direct transmission of maternal mtDNA to embryos through EVs.

A1. Thank you for your kind comments.

**Reviewer #2 (Public Review):**
Q2. In Bolumar, Moncayo-Arlandi et al. the authors explore whether endometrium-derived extracellular vesicles contribute DNA to embryos and therefore influence embryo metabolism and respiration. The manuscript combines techniques for isolating different populations of extracellular vesicles, DNA sequencing, embryo culture, and respiration assays performed on human endometrial samples and mouse embryos.Vesicle isolation is technically difficult and therefore collection from human samples is commendable. Also, the influence of maternally derived DNA on the bioenergetics of embryos is unknown and therefore novel. However, several experiments presented in the manuscript fail to reach statistical significance, likely due to the small sample sizes. This manuscript is a good but incomplete start as to the potential function of maternal DNA transfer via vesicles.In my opinion the manuscript supports the following of the authors' claims:1. Different amounts of nDNA and mtDNA are shed in human endometrial extracellular vesicles during different phases of the menstrual cycle.2. Endometrial microvesicles are more enriched for mitochondrial DNA sequences compared to other types of vesicles present in the human samples.3. Fluorescently labelled DNA from extracellular vesicles derived from an endometrial adenocarcinoma cell line can be incorporated into hatched mouse embryos.4. Culture of mouse embryos with endometrial extracellular vesicles can influence embryo respiration and the effect is greater when cultured with isolated exosomes compared to other isolated microvesicles.My main concerns with the manuscript:1. Several experiments presented fail to reach statistical significance or are qualitative.2. The definitive experiments presented in the manuscript are limited to the transfer of DNA in general not mtDNA. Therefore a strong connection with metabolism is missing, diminishing the significance of the findings.

A2. We thank you for your detailed feedback. While we acknowledge the reviewer's concerns regarding sample sizes, we emphasize that this study was intentionally designed as a pilot study and was approved by the IRB with a specific sample size to serve as proof of concept. We fully agree that further research is essential for a more comprehensive understanding of the novel biological process described in this manuscript. When this manuscript is finally accepted, we can submit a new IRB application to obtain a larger sample size, allowing us to delve deeper into demonstrating the connection with metabolism

Recommendations for the authors:
**Reviewer #1 (Recommendations For The Authors):**
Q3. The authors have made significant improvements, and the manuscript now is appropriate for eLife.

A3. Thank you for your consideration.

**Reviewer #2 (Recommendations For The Authors):**
The authors have made several changes that have improved the manuscript. However, I still have some concerns.Q4. The title is still too definitive. Something like "Vertical transmission of maternal DNA through extracellular vesicles is associated with changes in embryo bioenergetics during the periconception period" would be more appropriate.

A4. As mentioned earlier in the response to the editors, we acknowledge the concerns regarding the manuscript's title.

Following your recommendation, the proposed new title of the manuscript is “Vertical transmission of maternal DNA through extracellular vesicles associates with altered embryo bioenergetics during the periconception period.”

Q5. I am confused by the incorporation of the new experiment (supplementary figure 7) where embryos are cultured in free-floating synthesized mtDNA. If these sequences were not encapsulated in vesicles I don't think the experiment is relevant. If they were similarly prepared as in the section "Tagged-DNA production and EV internalization by murine embryos" I stand corrected but please clarify or omit. Otherwise, the new data/figure in response to Q11 showing co-localization of mitochondria and EdU-tagged DNA from MVs from Ishikawa cells is more compelling. However, this doesn't separate the uptake of mtDNA alone from the potential uptake of mitochondria, which this manuscript is not focused on.

A5. We apologize for any confusion that may have arisen for the reviewer. We conducted this experiment in response to question Q4 posed by the same reviewer, which specifically inquired about the detection of internalized mtDNA by the embryos.

As previously stated in the revised manuscript, the EdU system does not selectively label mtDNA; instead, it labels any newly synthesized DNA, both nuclear and mitochondrial. We have not found a system that specifically labels mtDNA for subsequent tracing inside EVs or for encapsulation within artificial EVs (which falls outside our expertise). Therefore, we employed labeled mtDNA that we could trace after the embryos' internalization.

While we acknowledge that this approach is not perfect, it does demonstrate the internalization of mtDNA sequences within the embryo. We have revised the manuscript to eliminate any potential sources of confusion. If the reviewer or editors still have concerns about the experiment's suitability, we are open to removing it from the final version of the manuscript. Please refer to page 9 and lines 234-238 for more details."